# Reinforcement learning when your life depends on it: A neuro-economic theory of learning

Jiamu Jiang[1], Emilie Foyard[1], Mark C. W. van Rossum[1,2] *

**1** School of Mathematical Sciences, University of Nottingham, Nottingham, United Kingdom, **2** School of Psychology, University of Nottingham, Nottingham, United Kingdom

* mark.vanrossum@nottingham.ac.uk

**Data Availability Statement:** Code is available at https://github.com/vanrossumlab/neuroeconomicRL.

## Abstract

Synaptic plasticity enables animals to adapt to their environment, but memory formation can require a substantial amount of metabolic energy, potentially impairing survival. Hence, a neuro-economic dilemma arises whether learning is a profitable investment or not, and the brain must therefore judiciously regulate learning. Indeed, in experiments it was observed that during starvation, Drosophila suppress formation of energy-intensive aversive memories. Here we include energy considerations in a reinforcement learning framework. Simulated flies learned to avoid noxious stimuli through synaptic plasticity in either the energy expensive long-term memory (LTM) pathway, or the decaying anesthesia-resistant memory (ARM) pathway. The objective of the flies is to maximize their lifespan, which is calculated with a hazard function. We find that strategies that switch between the LTM and ARM pathways, based on energy reserve and reward prediction error, prolong lifespan. Our study highlights the significance of energy-regulation of memory pathways and dopaminergic control for adaptive learning and survival. It might also benefit engineering applications of reinforcement learning under resources constraints.

## Author summary

There is increasing evidence that biological learning and in particular the creation of long lasting forms of memory requires substantial amounts of energy. It has been observed that as a result, animals such as drosophila might stop some forms of learning when they are low on energy. In this modelling paper we analyze this learning vs starvation trade-off using a hazard framework with as objective to maximize the lifetime of the animal. We then explore the optimal algorithm to balance energy saving with learning. We find that it is best to restrict the learning using expensive persistent memory to situations where the animal's energy reserve is high, and there is also a large deviation between expected and actual reward. We speculate that there is evidence for similar energy adaptive mechanism in mammalian learning. The findings might also be relevant for human behavior and artificial systems with resource limitations such as limited battery life.

**Funding:** JJ was supported by a Nottingham Vice Chancellor's International Scholarship, and Turing Enrichment Scheme R00700. The funders had no role in study design, data collection and analysis, decision to publish, or preparation of the manuscript.

**Competing interests:** The authors have declared that no competing interests exist.

## Introduction

Learning allows animals to adapt to their surroundings, evade dangers, and enhance survival prospects. However, learning itself comes at a cost as it requires considerable amounts of metabolic energy. For instance, experiments have shown that fruit flies that learn a classical conditioning task perish 20% faster when subsequently starved compared to starved control flies [1]. When they are not starved, flies strongly increase their food intake after learning [2, 3].

In Drosophila memory is expressed in (at least) two distinct pathways, that are believed to be mutually exclusive [4]. The Long Term Memory (LTM) pathway requires a lot of energy but yields persistent memory. Conversely, the Anesthesia Resistant Memory (ARM) pathway is thought to require negligible amounts of energy, as its expression does not significantly affect lifetime [1]. However, ARM memory typically dissipates within four days [5]. Whether ARM or LTM is expressed depends partly of the stimulus protocol, but also on the energy reserve of the animal. While it is interesting to speculate that the dependence on protocol is functional beneficial, e.g. [6], we solely focus on the role of energy reserve here. Notably, in aversive conditioning protocols flies halt energy-demanding LTM formation when starved [7].

As learning comes at a cost, a neuro-economic dilemma arises whether learning is a profitable investment or not. Yet, the energy requirements of learning have thus far been mostly overlooked in the computational community. The situation can be compared to the human dilemma whether or not to spend money on education: typically investment in education will pay off financially, but only if the life expectancy is long enough and bankruptcy can be avoided.

Here we examine the energy cost-benefit of learning on expected survival, and compare learning strategies that maximize survival during an aversive conditioning protocol. We introduce a hazard framework to examine the trade-off between the energy expenditure required learning and encountering hazardous stimuli. Learning to evade aversive stimuli decreases the stimulus hazard, but the energy expenditure associated with learning increases the starvation hazard. The objective for the flies is to maximize their lifetime by employing either the LTM or the ARM memory pathways. We propose a strategy that switches between ARM and LTM pathways depending on the current energy reserve and the reward prediction error. This strategy robustly increases lifetime across a number of stimulus protocols.

## Model: Hazard framework

Most biological reinforcement learning studies assume that animals seek to maximize total reward and minimize punishment. The tacit assumption is that this improves biological fitness. It is then common to compare behavior to reward maximizing policies, e.g. [8], often without regards for metabolic cost of implementing and updating the policy. Here, however, we directly assume that the optimal policy maximizes survival, i.e. the lifetime of the organism. Because learning requires energy, the policy needs to balance avoidance of a hazardous stimulus against expending of energy on learning.

To examine this trade-off we use a hazard function approach. Hazard functions were originally developed in life insurance to calculate the probability that policy holders would die; they are also used in failure analysis and healthcare, e.g. [9, 10]. In computational neuroscience hazard functions have been used to model the probability that a neuron fires a spike [11]. Despite being a natural approach, a hazard framework has to our knowledge not been used before for reinforcement learning problems.

Using a discrete time formulation, the hazard function $h(t)$ ($0 \leq h(t) \leq 1$) specifies at any time the probability to die within a time unit. The probability to have a lifetime $t$ is given by the probability of surviving all previous time-steps and perishing at time $t$, see Fig 1a. For a

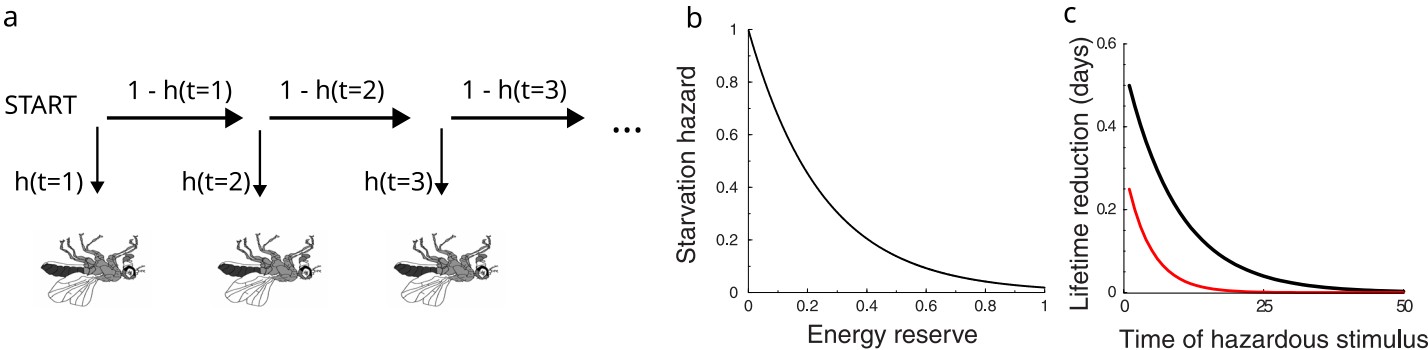

**Fig 1. Hazard framework.** A: Illustration of hazard formulation. At each day the fly has a probability $h(t)$ to die, or to survive to the next day. The hazard is determined by the fly's metabolic energy reserve and the stimuli it encounters. The hazard has two components: starvation hazard and hazard from approaching the noxious stimulus. B: Assumed relation between the normalized energy reserve of the fly and its starvation hazard. The hazard increases exponentially at low energy. Note that even at maximal energy, there is a background hazard C: Hazard framework leads to discounting. The reduction in lifetime due to an additional hazard versus the time of this extra hazard. Future hazards are exponentially less important than immediate ones. When the expected lifetime is shorter (red curve), the discounting is stronger, i.e. decay is faster. Baseline hazard: 0.1 (black), and 0.2 (red); hazard of stimulus in both cases 0.05. [fly image derived from https://commons.wikimedia.org/wiki/File:Hoxgenesoffruitfly.png. Bstlee, Public domain, via Wikimedia Commons].

constant hazard $h(t) = h > 0$, one finds that $P(t) = (1 - h)^{t-1} h$. The lifetime distribution is in this case exponential with mean lifetime $\langle t \rangle = \sum_{t=0}^{\infty} tP(t) = 1/h - 1$. For time-varying hazards, the probability to have a lifetime $t$ is $P(t) = S(t) - S(t + 1)$, where $S(t) = \prod_{t'=0}^{t-1} [1 - h(t')]$ is the survival function to survive until time $t$. The mean lifetime follows as

$$\langle t \rangle = \sum_{t=0}^{\infty} S(t) - 1. \tag{1}$$

In the following we measure time in days, and so the hazards have units 'per day'.

The total hazard can include factors such as the internal state of the animal, as well as external stimuli and environmental factors. We consider two hazards: First is the hazard from starvation, which increases when the metabolic energy reserve $M(t)$ diminishes. We assume that the energy reserve $M(t)$ is positive and saturates at 1, corresponding to about 1 Joule [12]. Although it would be straightforward to determine dependence of hazard on energy reserve experimentally, we are not aware of such experiments. Therefore we assume a steep increase at low energy levels, Fig 1b,

$$h_M(t) = \exp[-cM(t)] \tag{2}$$

We calibrate $c$ by using that well-fed flies ($M = 1$) have a lifespan of some 50 days [13], i.e. $c = 3.9$. Note that the hazard formulation includes the case where flies only die when the energy reaches zero. Hereto one would set the hazard $h_M(t)$ to a small background value whenever $M > 0$, and to one otherwise. The effect of such a hazard is shown in Fig A in S1 Text.

Second, there is a hazard associated to approaching the aversive stimulus. Although laboratory experiments generally involve non-lethal shock stimuli, in a natural environment such shocks could potentially forebode a life threatening event, for instance the presence of a predator. We denote this as the stimulus hazard $h_s^0$. With learning however, the animal will start to avoid the hazard. The resulting hazard is denoted $h_s(t)$ and is either $h_s^0$ or 0 (when avoided).

Being probabilities, hazards from different sources add up as $h_\Sigma(t) = 1 - [1 - h_s(t)][1 - h_M(t)]$. (In the limit of small $h_i$ or, equivalently, the continuum limit, this reduces to a regular sum.)

Interestingly, the hazard framework automatically leads to reward discounting—a core feature added by hand to many reinforcement learning (RL) models to express that immediate rewards are preferable to future rewards. In the hazard formulation rewards and hazards that are far in the future will hardly impact the lifetime. Instead it is important to minimize hazards early on. To illustrate discounting in a simple scenario, assume a constant permanent hazard and that at a certain time an additional hazard is introduced, active during one time-step only. The lifetime is reduced most if the hazard occurs immediately, whereas stimuli far in the future have no effect on the lifetime, Fig 1c. For a constant background hazard, the discounting can be shown to be exponential. Furthermore, when the energy reserve is low and the expected lifetime shorter, the discounting is stronger, Fig 1c (red curve). Thus discounting emerges automatically and unlike traditional RL does not require an additional parameter.

Hazard typically increases with age, however we assume that the experiments are so drastic that age dependence of the hazard can be ignored ("biologically immortality") or averages out. In more detailed models such effects could be included, and should find that expensive LTM learning is less beneficial for aged animals with little expected lifetime left, but would benefit young flies.

## Model: Network design

We implemented a network reflecting the Drosophila brain's anatomical structure, and a complementary feedback network associated with reinforcement, Fig 2. In Drosophila aversive conditioning experiments, an odor (conditioned stimulus, CS) is paired with a shock (unconditioned stimulus, US). By repeating exposure to the CS-US pairs a few times, the flies learn to avoid the odor, as can be subsequently tested in a T-maze. We model the case where an odor is presented each day, which when approached, leads to a noxious stimulus and hence is to be avoided.

The underlying circuitry, involving sensory encoding Kenyon Cells (KCs) and action-driving Mushroom Body Output Neurons (MBONs), is relatively well understood [5, 14, 15]. The network comprises a population of sensory KCs that represent the odor signal, which

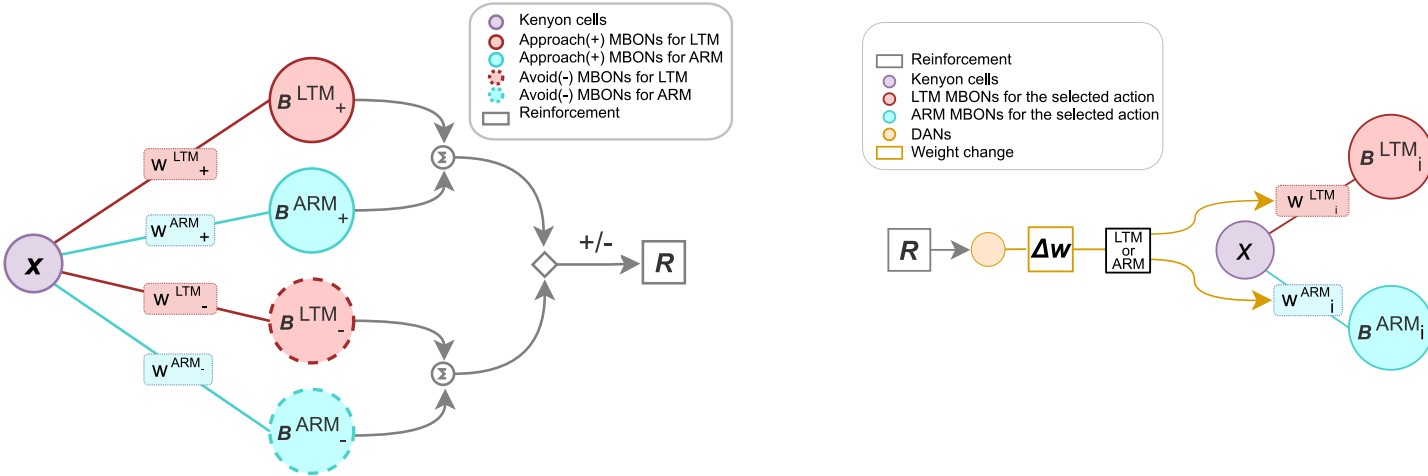

**Fig 2. Schematic of the learning network rooted in the Drosophila brain anatomy.** The left panel demonstrates the feed-forward Decision-making network. The $x$ indicates the input activity from the Kenyon cells, $w$ the synaptic weights, and $B$s are the MBON activities. The network is complemented by the right panel showing the adaptive learning mechanism, steered by reinforcement signals. The weight change, denoted by $\Delta w$, is modulated by the reinforcement. The synaptic weight changes are expressed in either the ARM or the LTM pathway. The subscript represents the action of the current trial, either approach (+) or avoid (−).

subsequently drive the Mushroom Body Output Neurons (MBONs) that determine behavior, Fig 2 left. The firing rate of the KC population is denoted $x$.

The activities of the MBONs are split up in the ARM and LTM pathways (see below). Each pathway is modeled as a linear neuron. Denoting their activities as $B$, we have $B_{\pm}^{LTM} = w_{\pm}^{LTM}x$, and $B_{\pm}^{ARM} = w_{\pm}^{ARM}x$, where $\pm$ indicates approach (+) and avoidance (−) behaviors, and the parameters $w_{\pm}^{LTM}$ and $w_{\pm}^{ARM}$ denote the synaptic strengths from the KCs to the MBONs. Given the additive nature of MBON signals [16], we posit that total neuronal activity driving the approach and avoidance behaviors results from the sum of the ARM and LTM components. Hence

$$B_{\pm} = (w_{\pm}^{ARM} + w_{\pm}^{LTM})x \tag{3}$$

The total weight for approach and avoidance behaviors is $w_{\pm} = w_{\pm}^{ARM} + w_{\pm}^{LTM}$.

Winner-Take-All competition between the two MBON neuron populations determines the fly's action. The competition process is not explicitly modeled, but could reflect lateral inhibition and attractor dynamics. We assume that the neural processing and resulting decision making is noisy. (Otherwise, even the smallest imbalance would fully determine the decision). This randomness also means that the organism does not fully commit to avoiding even the smallest hazard, but keeps exploring as well. Assuming independent Poisson spike-time variability, the input to the decision making neurons has a variance equal to the mean input. At sufficient high rates this is well approximated by normal distribution with a variance equal to the mean. The probability to avoid $P_-$ is a sigmoidal function of the difference in activities $B_+$ and $B_-$

$$\begin{aligned} P_- &= P(B_- > B_+) \\ &= \frac{1}{2} + \frac{1}{2}\mathrm{erf}\left(\sqrt{\mu}\,\frac{w_- - w_+}{\sqrt{w_-^2 + w_+^2}}\right) \end{aligned} \tag{4}$$

The mean $\mu$ of $x$ can be extracted from the observation that when learning is saturated the performance corresponds to about $P_- = 0.925$ [5]. Using $w_- = 1$ and $w_+ = 1/2$ (see below), this yields $\mu = 10.3$. The $\mu$ is the average number of spikes the MBON neuron receives from the sensory neurons within one integration period (e.g. counting spike of a 103Hz train during 100ms); encouragingly it is similar to the value used in [15].

## Reward driven plasticity

The reward when approaching (+) the aversive stimulus is negative and denoted $R_+$, without loss of generality we set $h_s = -R_+$. That is, the punishment is expressed as its hazard. The reward for avoiding the stimulus, $R_-$, is set to 0. In the MB of Drosophila, reinforcement-related signals are encoded by dopamine neurons (DANs) [17, 18], and these DAN signals modulate the plasticity of the synapse connecting KCs to MBONs [15, 19–21]. The synaptic strength associated with the selected behavior is updated based on the discrepancy between the reward from the current trial $R_{\pm}(t)$ and the expected reward $\bar{R}_{\pm}$, also known as the reward prediction error. The synaptic weight modification is

$$\Delta w_{\pm} = \eta[R_{\pm}(t) - \bar{R}_{\pm}(t-1)]x, \tag{5}$$

where $\eta$ is the learning rate. In line with experiments [21], the learning according to Eq 5, occurs through depression of the approach action, rather than a strengthening of the avoid action.

The learning rate was calibrated by using that in [5] after a single cycle of learning, avoidance performance was $P_- = 0.85$, which corresponds to $w_- = 0.8$. Using $h_s = 0.1$ we find $\eta = 0.6$. As in these experiments the performance early after learning through LTM and ARM is similar, the same learning rate was used for both ARM and LTM learning. A lower ARM learning rate, reduces the benefit of ARM. Under the M1 energy model (see below) a reduction of LTM learning rate leads to weaker learning, but energy expenditure is less. As a result, a reduced LTM learning rate somewhat increases lifetime at low energy reserves but decreases it at high reserves, Fig B in S1 Text.

The $\bar{R}_\pm$ in Eq 5 is the running average of the reward of either action. The expected rewards are initialized at zero at the beginning of the simulation. The expected reward is updated when that action is chosen, otherwise it decays to zero

$$\bar{R}_\pm(t) = (1 - \alpha)\bar{R}_\pm(t) \qquad \text{if not choosen} \tag{6}$$

$$= (1 - \alpha)\bar{R}_\pm(t) + \alpha R(t) \qquad \text{if choosen} \tag{7}$$

where $\alpha = 1 - e^{-1/\tau_R}$, and the decay time constant of the average, $\tau_R$, is set equal to the ARM decay (below). In our simulations the value of $R_-$ and $\bar{R}_-$ (the reward for avoiding) remain zero throughout and could in fact be omitted all together, but are included for generality. When extending to multiple odor associations, each stimulus would carry it's own expected reward. Finally, it would also be of interest to study other decay dynamics, or an implementation via network interactions [15].

## Synaptic plasticity pathways

Experiments show that ARM and LTM memory formation are mutual exclusive [4]. Hence the synaptic weight changes given by Eq 5 are expressed in either LTM or ARM weights. Updating the weight in the ARM pathway ($w_\pm^{ARM}$) comes at negligible metabolic cost [1, 2]. However, the ARM weights decay over time, so that the update equation reads

$$w_\pm^{ARM}(t) = \gamma_{ARM} w_\pm^{ARM}(t - 1) + \Delta w_\pm \tag{8}$$

Here $\gamma_{ARM}$ is the ARM decay rate. To estimate its value, we use the data in [5], where flies where exposed to massed training and the memory decay was measured. In four days the probability for the correct action decayed from $P_- = 0.925$ to $P_- = 0.525$ (in terms of the performance index used there, from 85% to 5%). In the model the memory extinction is found by substitution of Eq 8 in Eq 4. A fit yields $\gamma_{ARM} = 0.34$.

When, in contrast, LTM is expressed, the weight updates do not decay

$$w_\pm^{LTM}(t) = w_\pm^{LTM}(t - 1) + \Delta w_\pm \tag{9}$$

However, LTM is metabolically costly [1]. We examine two abstract energy models. The first assumes that the metabolic energy cost of LTM formation decreases the energy reserve by an amount proportional to the weight change [22]

$$M_1(t) = M_1(t - 1) - c_{LTM}(|\Delta w_+^{LTM}| + |\Delta w_-^{LTM}|) \tag{10}$$

The parameter $c_{LTM}$ denotes the energy cost of LTM. In experiments LTM before starvation reduced the survival time in female flies from 26 to 22 hrs [1], this approximately corresponds to $c_{LTM} = 0.27$, see [12] for details. A larger value would lead to more costly LTM and would shift the hazard at which LTM becomes beneficial to larger values.

At the start of the simulation the combined weight needs to be different from zero. Furthermore, for computational convenience, the ARM decays to zero. As a results the ARM weights were initialized at 0; the LTM weights at 0.5.

An alternative energy model, termed $M_0$, assumes that energy is used whenever LTM plasticity occurs, but the amount is independent of the amount of synaptic strength change,

$$M_0(t) = M_0(t-1) - d_{LTM} \qquad \text{if } |\Delta w_{\pm}^{LTM}| \neq 0.$$

Simulation of the single exposure experiments, yields a calibration $d_{LTM} = 0.1$. Mathematically, the energy models corresponds to the $L_1$ and $L_0$ norms of the weight updates [23]. The subscript distinguishes between the two variants for the energy used by LTM. To summarize

$M_1$-**energy**: Energy equals the total amount of LTM synaptic weight change, e.g. number of receptors inserted and removed.

$M_0$-**energy**: Energy equals the total number of LTM events.

We are not aware of experiments that decide between these energy models; future experiments hopefully will. Note that interactions between ARM and LTM pathways as well as interactions across time, that could in principle increase or reduce energy requirements, are also ignored. Furthermore, the dynamics of the synaptic weights are modeled as first order equations, hence phenomena like delayed expression of LTM [24] are not included. Finally, the model is agnostic about other forms of memory, such as STM; in that sense ARM stands here for any metabolically inexpensive, decaying form of memory.

## Stimulus protocol

In the simulation an odor is presented each day, which when approached, leads to a hazard of killing the fly and hence is to be avoided. In detail, on every day: the fly chooses stochastically to approach or avoid the stimulus (Eq 4); the reward and reward expectation are updated (Eq 7); the synapses are updated (Eq 5); the energy reserve is updated; the hazards are calculated; and finally the expected reward and ARM weights are decayed. The protocol is given to 10000 flies and repeated 50 days. This is much longer than the average lifetime and ensures that one can be certain that all flies will have died by then.

The simulation contains in principle two stochastic elements: first, the decision to avoid the stimulus stochastic (Eq 4) and, second, the hazard is a probability to be evaluated every day for every fly, Fig 1. As a technicality, by calculating the population average expected lifetime from the hazards (Eq 1), we remove this second source of variability in the simulation and reduce variability that would otherwise require larger simulated populations. Code for the paper can be found at github.com/vanrossumlab/neuroeconomicRL.

## Results

### ARM versus LTM learning

We first illustrate the model by assuming that flies exclusively use either the ARM or the LTM pathway. We simulate a population of flies that is subject to the following experiment: Each day an odor is presented, which when approached, leads to a hazard of killing the fly and hence is to be avoided. In addition there is a hazard to die from starvation, Eq 2. We initially assume that apart from the energy required for LTM learning, there is no change in the energy reserve in the flies.

We track the evolution of the hazard, synaptic weights, energy reserve, and performance as measured by avoidance of the stimulus, Fig 3a. With ARM-pathway learning (left panels),

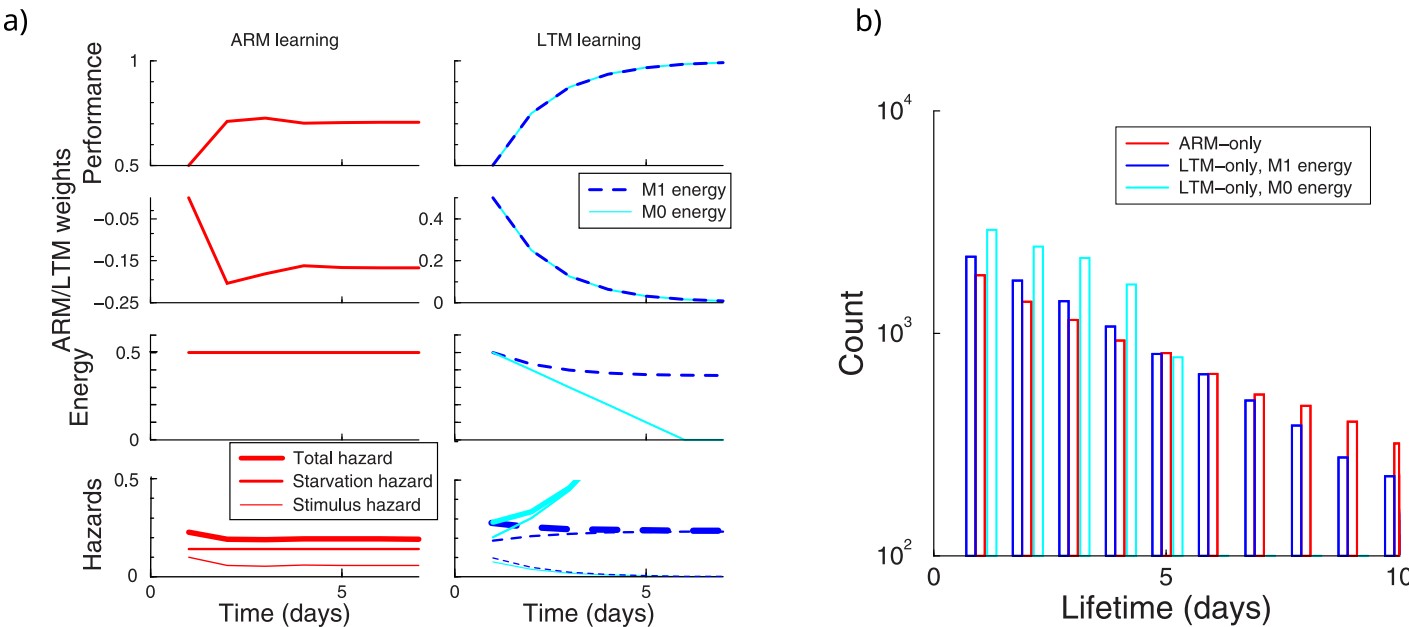

**Fig 3. ARM vs LTM learning during the simulated aversive conditioning protocol.** A: Evolution of performance, weights, energy reserve, and hazard. Left panel: ARM only learning. Right: LTM learning under the $M_0$ and $M_1$ energy model. B: Lifetime histogram of 10000 flies under either pathway. Because the total hazard variations are relatively small, the distributions are close to exponential. However, for the $M_0$ energy model, lifetimes are much shorter. (Parameters: stimulus hazard $h_s$ = 0.2, initial energy reserve 0.5).

performance improves over the days but does not exceed 70%, as the flies forget between the exposures. As a result the hazard from stimulus exposure (the product of the hazard itself and the chance of encountering it) remains substantial. However, the energy reserve stays high and starvation hazard low. The synaptic weights (and as a result behavior) oscillate slightly before settling down due to the updates in the expected reward.

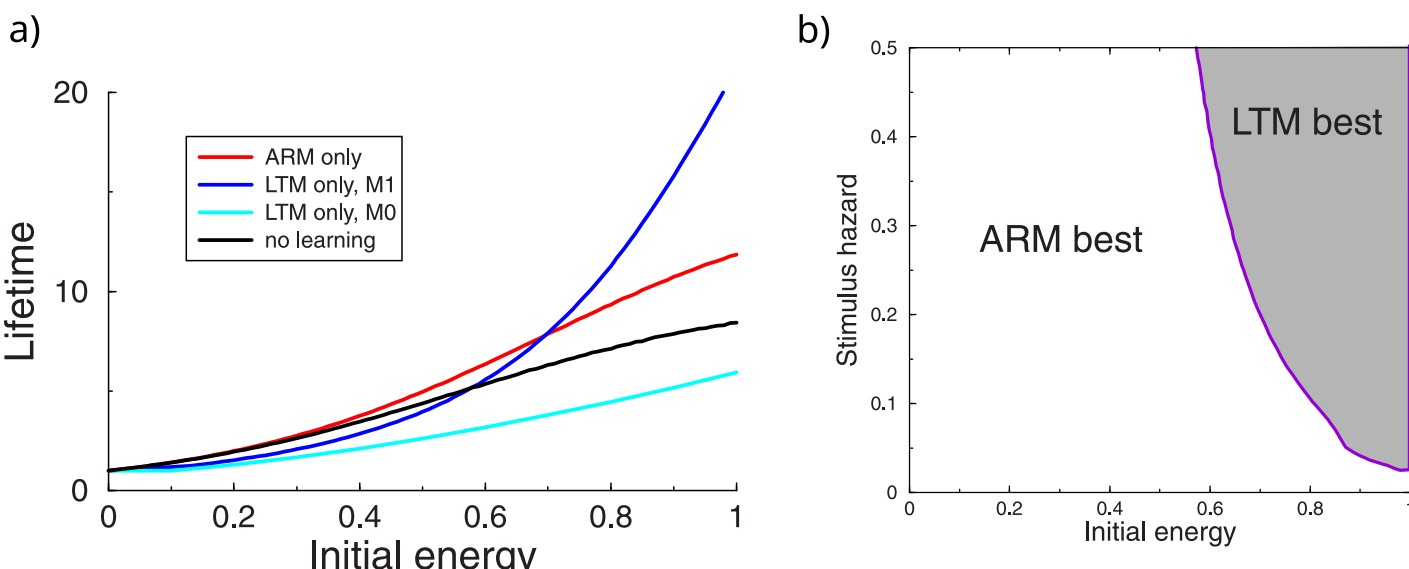

**Fig 4. Effect of ARM and LTM learning on lifetime.** A: Lifetime as a function of the energy reserve at day 0. ARM learning (red curve) is always better than no learning (black line). LTM (blue curve) is only beneficial when the energy reserve is high and the energy use is proportional to update size. (Stimulus hazard $h_s$ = 0.1). B: Both the initial energy and hazard level influence whether LTM learning increases lifetime over ARM learning. When the hazard is high, is better to invoke LTM at lower energy reserves ($M_1$ energy model).

In contrast, in LTM learning performance becomes close to perfect after some 4 days, always avoiding the stimulus hazard, Fig 3a right panels. The population performance grows smoothly, because on the first trial only half the flies will randomly approach the stimulus and will learn, and so on.

While the hazard will be avoided, the expenditure of energy needed for LTM learning increases the starvation hazard. This effect is mild if the energy used by LTM is proportional to the size of the weight update ($M_1$ energy). In this case the difference between the reward and its expectation and hence the amount of weight change diminishes as learning progresses. Only the first few learning events are costly (blue curves). However, when the cost is independent of the amount of weight update ($M_0$, cyan curves), the energy is quickly depleted and the starvation hazard rises rapidly.

In this example ARM learning yields the longest lifetime of 5.6 days, LTM learning yields 4.4 days using the $M_1$ energy model; their lifetime distributions are close to exponential. Using the $M_0$ energy model the lifetime is only 2.4 days, Fig 3b.

These simulations raise the question which memory pathway generally yields to longest lifetime for a given hazard and initial energy reserve. We varied the initial energy reserve of the flies, and determine the lifetime with ARM and LTM learning and in the absence of learning, Fig 4a. Because in the model ARM learning (red curve) comes at no cost, it is always better to learn with ARM than not learning at all (black curve). Under the $M_0$ energy model, LTM learning never extends lifetimes (the cyan curve lies under all others). Under the $M_1$ energy model (blue curve), there is a transition point. When initial energy is low, avoiding starvation is more important than avoiding the hazard, hence ARM yields longer lifetimes than LTM. In contrast, with a large energy reserve, the investment in avoiding the hazard is worthwhile and LTM yields a longer lifetime. The point at which LTM is better, depends on the stimulus strength. The higher the hazard, the lower the transition point, Fig 4b. In the Appendix (Suppl. Material) we derive an equation that gives insight in the break-even point, however, a full analytical treatment seems out of reach, because learning does not only affect the next decision, but all future (discounted) decisions. In the remainder we therefor rely on simulations.

The role of the stimulus interval is subtle, Fig C in S1 Text. ARM memory decays more when the interval is longer, but also the overal stimulus hazard decreases (in the extreme case that a stimulus never repeats, LTM is only detrimental). As a result, both ARM and LTM are less beneficial.

## Threshold models

In the above simulations the memory pathway was set once and for all at the start of the simulation. While this is useful to gain understanding, it makes more sense to choose the pathway depending on the *current* energy reserve $M(t)$. We assume that the expensive LTM pathway was used whenever the energy reserve exceeded a threshold, otherwise the ARM pathway was updated. To show the benefit of this algorithm we consider a population of flies with different initial energies, drawn uniformly between 0 and the maximum and measured the average lifetime as the threshold was varied, Fig 5. Note that, as expected, when the threshold is 0 (1), the lifetime equals that of LTM-only (ARM-only). When the threshold parameter is tuned (x-axis), the lifetime can exceed that of exclusively using either LTM (blue) or ARM (red). The peak of the curve shifts left as the stimulus hazard increases. That is, the larger the stimulus hazard, the lower the threshold that maximizes lifetime. Under the $M_0$ energy model, adaptive learning is only beneficial for large stimulus hazards, Fig 5 bottom row.

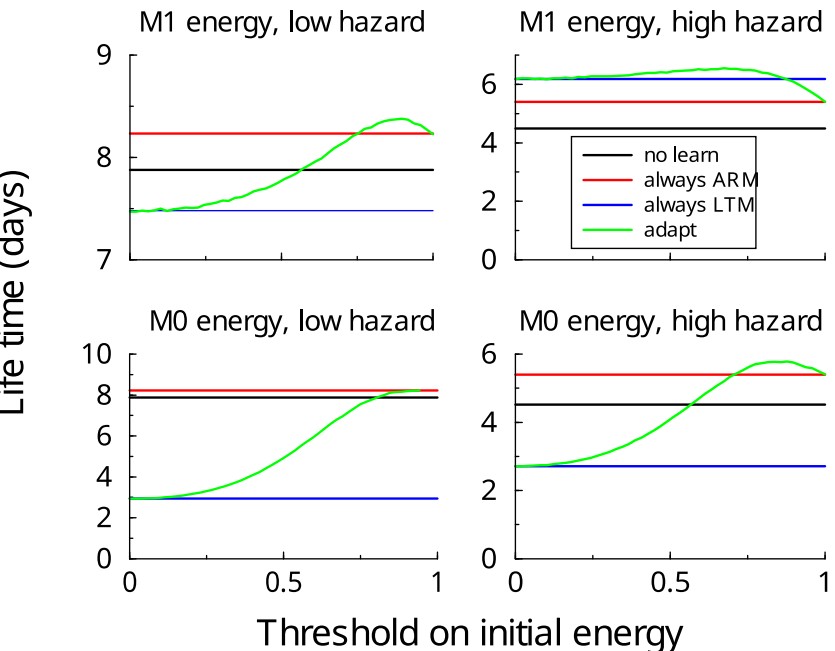

**Fig 5. Adaptive switching between ARM and LTM can improve lifetime.** Population lifetime vs threshold for two stimulus hazard levels. LTM was employed whenever the energy exceeded a certain threshold (x-axis). Left: low (0.05) and high stimulus hazard (0.2). Using the $M_1$ energy, the adaptive model (green) increases average population lifetime compared to either LTM or ARM exclusively. For the $M_0$ energy model, adaptive learning is only beneficial at large hazards. The optimal threshold that gives the longest lifetime depends on stimulus hazard.

## General threshold models

In the above adaptive switching model, LTM will be employed when the energy reserve is sufficient, even if the reward prediction error and hence weight changes are small. This means that energy might be spend for only a small change in avoidance behavior. Therefore, we made the threshold both dependent on the current energy reserve $M(t)$, and the difference between expected and actual reward, $\Delta R = |R(t) - \bar{R}(t-1)|$. We parameterized the switch so that the LTM pathway was employed whenever

$$c_M M + c_R \, \Delta R > 1$$

The parameters $c_M$ and $c_R$ define a line in the $M, \Delta R$-plane. When $c_R$ is set to zero, we retrieve the energy threshold model: $M$ has to be larger than $1/c_M$ for LTM to occur, Fig 6a. Likewise, when $c_M = 0$ the decision solely depends on $\Delta R$. Provided that $c_M > 0$, a large reward prediction error $\Delta R$ will lower the threshold for LTM memory.

We varied the stimulus hazard and optimized the $c_M$ and $c_R$ parameters of the threshold. The lifetime is maximal around $c_M = 1.01$ and $c_R = 1.76$, Fig 6b. If a threshold on just the energy were optimal, the optimal threshold would be lying on the $c_R = 0$ axis. And similarly, when just a threshold on $\Delta R$ would suffice, the optimal solution would lie on on the $c_M = 0$ axis. As the optimum lies away from both axes, a joint threshold yields the longest lifetimes.

The lifetime using optimized parameters exceeds that of exclusively using the ARM or LTM pathway across stimulus hazards, Fig 6 right. The adaptive threshold model picks the 'best of both worlds'.

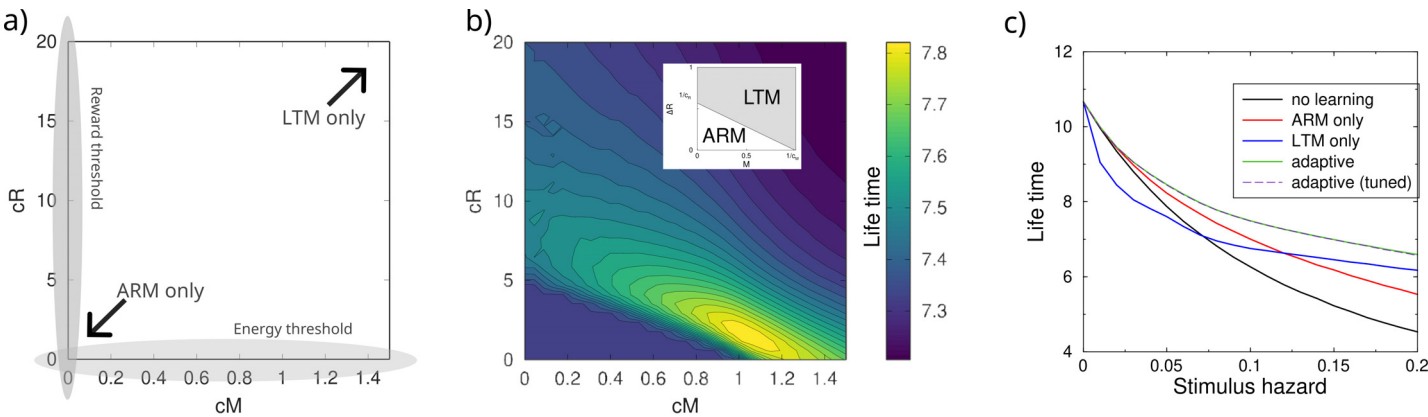

**Fig 6. Adaptive plasticity model with dependence on both the current energy reserve and the reward prediction error.** A: Schematic of the role of parameters $c_M$ and $c_R$. LTM was only used when $c_M M + c_R |\Delta R| \geq 1$. B: lifetime as a function of the threshold parameters. lifetime was averaged across stimulus hazards. Inset show the corresponding optimal threshold model. C: Lifetime as function of the stimulus hazard. The adaptive plasticity yields the longest lifetime. (M1 energy model).

Ideally, the optimal threshold should be such that a change in stimulus hazard should not require a re-tuning of the threshold parameters. We calculated the lifetime for the parameters that were best on average, and compared it to the lifetime optimized for each value of stimulus hazard. The lifetimes using the fixed parameters were practically indistinguishable from the individually tuned threshold parameters (overlapping top curves in Fig 6. Hence the threshold model is robust against changes of the stimulus parameter.

We also tried a variant in which either energy reserve *or* reward error where above a threshold, as well as a model in which both energy reserve *and* reward error needed to exceed a threshold; after optimization these performed as well as the above model, at least on the given task but not better, Fig D in S1 Text.

For the $M_0$ energy model the results are very comparable (Fig E in S1 Text). As above, under the $M_0$ energy model the lifetime is severely shortened when always using the LTM pathway, because every LTM plasticity is expensive even if the weight changes are small. But again, adaptively switching to LTM under the right circumstances improves lifetime. The optimal $c_M$ parameter is somewhat smaller ($c_M = 0.97$, $c_R = 2.35$), that is, the energy needs to be larger to switch to LTM than for the $M_1$ energy model, Fig E panel b in S1 Text.

We repeated this analysis for two other parameters of the stimulation protocol. First, we fixed the stimulus hazard (0.1), but we assumed that approaching the stimulus only sometimes lead to exposure to the hazard. The hazard probability was varied between 0 and 1, and determined on each trial independently whether the hazard was encountered or not. As expected, at the zero stimulus probability, the lifetime was maximal and independent of any learning. Again the adaptive threshold robustly improved lifetime, Fig 7 left.

Next, we modified that model so that in addition to the energy expenditure by LTM plasticity, there was a fixed daily energy intake/expenditure. The lifetime has a sigmoidal shape as a function of this amount, Fig 7 right. When there is a high expenditure (left part of graph), the fly heads for perishing anyway, and investment in LTM learning only hastens that (blue curve lies below red curve). But when there is a daily net intake, the investment in LTM memory helps to escape the hazard, while future starvation is unlikely. The lifetimes using LTM memory now exceed those from ARM learning. Again the adaptive algorithm improves the lifetime, outperforming either ARM or LTM exclusive learning. Also for the $M_0$ energy model adaptive learning improves lifetimes for both daily energy intake/expenditure and the probabilistic stimulus, Fig F in S1 Text.

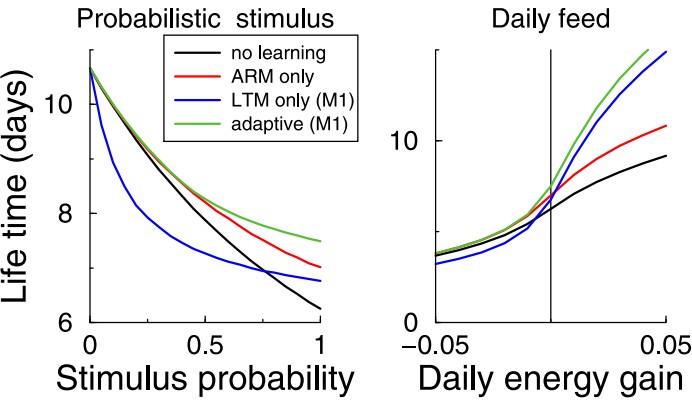

**Fig 7. Role of stimulus parameters.** Left: The lifetimes for a probabilistic stimulus against the probability of encountering the stimulus when approached. Right: Lifetime when there is additional daily energy intake or loss (stimulus probability is one). In both cases the adaptive algorithm robustly outperforms fixed strategies (Stimulus hazard 0.1; $M_1$ energy model).

## Appetitive conditioning

While we designed the model for avoidance conditioning, a similar circuit is thought to underlie appetitive conditioning. The transient memory pathway is in this case STM instead of ARM [24, 25]. We assume that like ARM, STM decays quickly and has a negligible metabolic cost. We assumed that approaching the reward increased the energy reserve with half the maximum reserve, i.e. 0.5 units (but the total reserve was still capped at one). In addition, the flies used a fixed amount of energy every day. The only hazard that the fly encountered was from starvation.

The lifetime under the ARM and LTM learning pathway is shown in Fig 8a as a function of the daily energy use. The daily use is the main determinant of the lifetime. For high daily use, lifetimes are short as the reward is not sufficient to prevent starvation, while for zero daily change there is no benefit in learning. However, in the intermediate region, learning increase lifetime. For daily stimulation (a. left panel), ARM learning is best. ARM learning performs very well because, in contrast to the aversive protocol, the ARM memory is daily refreshed by approaching the stimulus and boosted in appetitive conditioning. However, for longer intervals the benefit of ARM diminishes due to its decay(a. right panel).

It is known that flies also switch between LTM and short term memory pathways in appetitive conditioning. However, in contrast to aversive conditioning, the LTM pathway is only activated when the animals are starved prior to conditioning [25]. Unlike the experimental findings, we find that LTM is more beneficial at high energy than at low energy reserves, Fig 8b. When the animal has enough reserve there is no reason not to express LTM. However, it might be that LTM requires other resources that are scarce, or that LTM learning carries other detrimental consequences. Instead, in the model, the expenditure rate rather than the reserve is the critical factor. After all, LTM has less benefit when the expenditure is high, so that it is unlikely that the LTM can ever be assessed, nor when the expenditure is very small, so that there is no chance of starvation anyway.

## Discussion

Inspired by experimental findings that LTM memory formation is metabolically costly, and that flies stop aversive LTM learning under starvation, we have explored how such adaptive

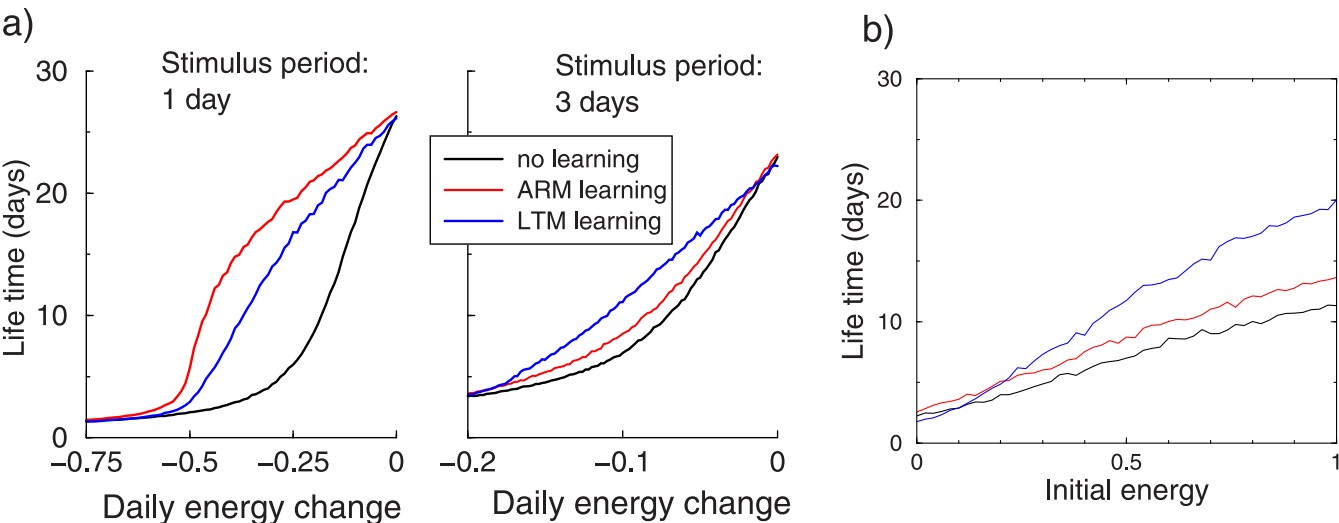

**Fig 8. Appetitive conditioning.** a) Lifetime as a function of daily energy use. Upon approaching the stimulus, the animals gain half a unit of energy. Right: as left panel but with 2 days between the stimuli. In this case LTM is better. The adaptive algorithm fully overlaps with the maximum of these curves (not shown for clarity). b) Lifetime as a function of initial energy reserve, in case of a 2-day stimulus interval and -0.1 daily energy change. Lifetime gains with LTM are highest when the reserve is high.

learning can increase evolutionary fitness and how the switch between LTM and ARM should be set. Using the hazard framework, a switch to LTM when the energy reserve is high and the reward prediction error is high, improves population lifetime.

Necessarily, simulations need to assume a certain hazard exposure protocol. The optimal parameters that set the switch point will be dependent on this. But some generalizations are immediately obvious. For instance, when the stimulus interval is increased, the ARM memory will decay more between events, and ARM becomes less effective. As a result the fly should switch to LTM sooner. As another extreme example, if the stimulus were only presented once, learning would be useless and should be turned off. The biological parameters have presumably been optimized for performance across the ensemble of naturally encountered environments and hence the parameters values found here are not expected to be exactly those found in experiments. Future studies could aim to close this gap and study more realistic and richer environments, including those with temporal correlations. The adaptive algorithm might be adjusted to include stimulus repetition and spacing effects [6].

It would also be of interest to include more complex memory dynamics that for instance include the slow rise of LTM expression [24], or 'hetero-synaptic' effects on the CS- pathway [26].

Another extension would be the learning of multiple associations. The current model assumes that each odor activates a distinct KC population, so that each association would cost extra energy. However, it would be straightforward to learn multiple associations that include overlapping activations.

We have relied on mean population lifetime as fitness measure, however true fitness is the ability to pass genetic material to offspring. A more involved model could use a fitness measure that reflects that. For instance, for a population it might be better to have a wide spread in the lifetime distribution, so that some individuals would survive periods of famine.

While the current work focused on Drosophila anatomy and physiology, there are indications that similar principles might be at work in mammals. In contrast to the fruit-fly's ARM

and LTM pathways, transient and persistent forms of mammalian long term potentiation (LTP) appear to be expressed at the same synapse. However, also in mammals there is physiological evidence for down-regulation of persistent LTP under energy scarcity via the AMPK pathway [27], and there is behavioral evidence for a correlation between blood glucose level and memory formation [28, 29]. Likewise, a dopamine reward signal, typically interpreted as signaling the reward prediction error, lowers the threshold for late-phase LTP [30–32].

Finally, reinforcement learning has many engineering and software applications. The results found here could potentially enhance the performance of RL algorithms, especially in resource-limited settings or tasks requiring multi-objective optimization. The energy requirements in these applications could be associated to computing the weight updates. Moreover also for computer hardware, memory storage is energetically expensive.

## Supporting information

**S1 Text. Single file with Appendix and 6 supplementary figures.**
(PDF)

## Acknowledgments

Discussions with Silviu Ungureanu, Long Tran-Thanh, Andrew Lin, and Walter Senn are gratefully acknowledged.

## Author Contributions

**Formal analysis:** Jiamu Jiang, Mark C. W. van Rossum.

**Investigation:** Jiamu Jiang, Emilie Foyard, Mark C. W. van Rossum.

**Software:** Jiamu Jiang, Emilie Foyard, Mark C. W. van Rossum.

**Supervision:** Mark C. W. van Rossum.

**Visualization:** Jiamu Jiang, Mark C. W. van Rossum.

**Writing – original draft:** Jiamu Jiang, Mark C. W. van Rossum.

**Writing – review & editing:** Emilie Foyard, Mark C. W. van Rossum.

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
