## [Decision Letter · Decision Letter 0]

24 Jun 2024

Dear Dr. van Rossum,

Thank you very much for submitting your manuscript "Reinforcement learning when your life depends on it: a neuro-economic theory of learning" for consideration at PLOS Computational Biology.

As with all papers reviewed by the journal, your manuscript was reviewed by members of the editorial board and by several independent reviewers. In light of the reviews (below this email), we would like to invite the resubmission of a significantly-revised version that takes into account the reviewers' comments.

We note that the reviewers were very positive about the approach and the potential for new insight it offers. However, they also suggest that the work could benefit significantly from a widened exploration the model e.g. in complex/realistic learning situations or in robustness to variance from assumed parameters. It would also be valuable to more directly explain the connection of the model conclusions to current biological understanding of the different memory mechanisms in Drosophila.

We cannot make any decision about publication until we have seen the revised manuscript and your response to the reviewers' comments. Your revised manuscript is also likely to be sent to reviewers for further evaluation.

Sincerely,

Barbara Webb

Academic Editor

PLOS Computational Biology

Daniele Marinazzo

Section Editor

PLOS Computational Biology

Reviewer's Responses to Questions

**Comments to the Authors:**

Reviewer #1: This study provides a reinforcement learning account of the presence of short-term and long-term memory systems under the assumption that long-term memory requires more resources. The objective to be optimized is the organism's lifespan, which is determined by its starvation state and dangerous stimuli.

Overall, I think this is an interesting and underexplored topic, and I think the model that the authors have written down is interesting. I think the paper would benefit from some additional exploration of the model and its dependence on key parameters, and I have some suggestions in this regard below.

Comments:

1) In general, this study provides an interesting and creative framework, but relatively simple investigations of that framework. I think the biggest question mark is how the model scales to multiple memories. Do the tradeoffs between LTM and ARM become more or less stark when there are multiple associations that the animal is confronted with? Does this depend on the time between stimulus presentations?

2) The variables in Fig. 2 should be defined in the caption.

3) The conclusions of the section on appetitive conditioning seem relatively weak. This seems likely to stem from an assumption that flies re-encounter the same stimulus every day. If a rewarding stimulus is encountered only sporadically, how will things change?

Reviewer #2: This manuscript reports an original and brilliant modeling study, tackling the question of memory regulation from a ‘neuro-economic’ perspective, i.e. taking into account not only the ‘information’ part of memory encoding, but also its metabolic cost. This approach is biologically relevant, as it is inspired by recent results obtained in the fruitfly Drosophila, where the metabolic cost of long-term memory formation and its implication in the regulation of memory dynamics has been clearly demonstrated experimentally.

In the model part, the authors explain their use of a hazard framework to estimate the survival probability of individual organisms (‘flies’). Hypotheses and the logic of the model are exposed in a very didactic way, so that the manuscript remains easy to follow even for readers that would not belong to the modeling field.

Across the results section, the model is first tested in a basic form (no adaptive plasticity) and gets increasing complexity, becoming more and more ‘realistic’ (random encountering of stimuli, adaptive switching between different memory modes, daily food intake).

- A memory with higher cost but longer persistence (LTM) can be more suitable to optimize lifetime than a memory with no cost but lower persistence (ARM), when available energy is high and hazard is strong.

- For a memory system which -like the fly brain- is capable of both ARM and LTM, the ability to switch between the two modes based on the energy reserve, or on a combination of energy reserve and reward prediction error is beneficial as compared to either type of memory alone.

These conclusions may seem intuitive when formulated in plain words. The strength of this work is to reach these conclusions based on few assumptions, which are firmly rooted in experimental evidence: (i) memories are encoded as changes in defined synaptic weights, reproducing the well-characterized architecture of the Drosophila mushroom body circuits; (ii) the metabolic cost of LTM is higher than that of ARM.

This work overall builds, to my knowledge, the first theoretical framework accounting for the metabolic control of memory dynamics, and for the existence of adaptive plasticity within memory systems. By showing that ‘simple’ arguments based on organism fitness can explain specific features of memory regulation, this work paves the way to an evolutionary-based and predictive modeling of memory regulation.

This study is therefore highly relevant, well-conducted, and far-reaching. The manuscript is pleasant to read, and deserves publication, provided the few concerns and questions expressed below are addressed.

1) In the few cases where some parameters or hypotheses of the model were determined arbitrarily, it would be interesting to give insight on how robust the results would be with respect to deviations from these choices. For example: how critical is the choice of an exponential law for the starvation hazard (eq.2)? What if the ARM and LTM learning rates are not assumed to be equal? How do the results vary (qualitatively or quantitatively) if the cost of LTM cLTM) is varied?

2) There are possibly some points of concern on Fig.3b, which seems in contradiction with the corresponding main text. The histogram on the figure indicates that the ARM-only condition gives a shorter lifetime than M1-LTM (given that there are more flies with longer lifetime in the M1-LTM condition), but the main text states the opposite (line 208). Also, it looks like the number of flies for the M0-LTM condition is larger than 10000 (the 4 bars are each >10^4). What is the explanation for that ? I might have misunderstood what is actually plotted, but if so I might not be the only one, and this figure would in that case deserve more explanation.

3) In the final part, the authors adapt the model to appetitive memory.

- A surprising result is that appetitive memory seems to have very little beneficial effect, long-term memory in particular being almost always detrimental as compared to no learning. Given that ‘real’ animals do make appetitive LTM, it could be advisable that the authors provide a more developed discussion about which features, specific to appetitive memory, their model fails to capture – or erroneously captures. Related to the previous point, could it be a wrong estimation of the energy gain associated with reward ingestion compared to the actual memory cost? Or of the efficiency of appetitive LTM formation, which is known to form after a single trial in starved flies?

- Could the authors elaborate more on the absence of effect of learning (even at zero metabolic cost) on lifetime in the regime of strong negative daily energy intake? As the initial energy level, and the value of the daily energy intake are strong determinant of the lifetime in the ‘no learning’ condition, this seems very counter-intuitive as learning would tend to increase the daily energy intake. I don’t see what in the model could be at the origin of this (lack of) effect.

- Also, could the authors detail on which basis it is speculated that ARM and LTM also exist in appetitive learning in flies?

- On Figure 8b, is the ‘daily energy gain’ including the amount of energy received by eating the reward (in that case in it different from the aversive case of Figure 7) or is it a fixed amount that is received independently of the behavioral choices?

In addition to these concerns, minor clarifications are needed:

- In Figure 7b, is the stimulus occurrence probabilistic as well (as on the left panel), or is it fixed as in the previous cases?

- L. 129: from what I understand, the R- parameter remains equal to zero at all times, and so does its expectation value “R-bar”. Do auhors confirm, and why is this parameter included in the model, then?

Finally, throughout the manuscript, there are quite a few typos/missing words/mistakes. I have listed hereafter all those I detected, but the authors should be encouraged to make a thorough proofreading of the manuscript upon revision.

l.24 : the publication showing increase in food intake is Plaçais et al., Nat. Communications (2017), rather than Plaçais & Preat (2013).

l.61 : It is surprising that the P(t) formula yields P=0 at all time for h=0 (i.e. no hazard). Do the authors confirm ?

l.83 : ‘RL’ is not defined. (Reinforcement learning ?)

Figure 5, top row : legend should be in blue for ‘always LTM’, not turquoise.

4th line of Figure 5 legend : typo at ‘compared’.

L 241 : missing word after ‘line’.

l.257 : ‘were’ instead of ‘where’ (I suppose).

l 298 : missing word after ‘might’

l.304 : ‘LTM’ already includes ‘memory’.

Reviewer #3: Overview: Jiang et al. present a computational study into reinforcement learning in Drosophila that incorporates the energy demands of aversive memory formation. Specifically, the authors extend the classic RL paradigm with two features: 1) that the objective is not to maximise reward, as in standard RL tasks, but to maximise lifetime, which itself depends on the fly’s energy reserves; 2) that learned memories are stored in one of two systems, which have different energy requirements and different lifetimes. This is an interesting new perspective for the neurosciences, and will likely encourage similar approaches to the study of memory dynamics, though the principle of maximising lifetime in an RL context has been described in other domains. The manuscript is well written and enjoyable to read, though several grammatical errors (including missing words) occur throughout, so need going through with a fine comb. There are some important modelling assumptions, described in more detail below, that would benefit from further clarification and discussion, as there are several discrepancies between the assumptions and experimental evidence.

Priority comments:

The model builds upon several assumptions that make it difficult to apply the principle ideas and results in this manuscript to long-term memory dynamics in Drosophila, and this is especially important for readers who may be naive to the extensive literature on learning in Drosophila. The assumptions that seem problematic are 1) that the same stimulus-reinforcement exposure can induce either ARM or LTM; 2) that ARM or LTM formation depend only on energy reserves, and not on each other; 3) ARM and LTM have the same learning rate. I go into further details for each point below, but I don’t think these details necessarily preclude the importance of the main ideas explored in this manuscript. However, I do think the authors’ ideas need to be presented in a different way in order to appropriately distinguish the concepts addressed in the model, and the relevance/application of those concepts to memory formation in animals, particularly Drosophila. I see a couple of potential options for going forward: 1) if there is good experimental evidence for it, the authors could reframe their presentation, focusing on different levels of memory consolidation (e.g. MTM and ARM, or STM and MTM etc.) that do not require such different induction protocols as do ARM and LTM; 2) the authors discuss the caveats of their model in greater depth, and explicitly describe the ways in which their model does and does not correspond to the experimental literature. Such a discussion should cover the additional details provided in the following.

1) The formation of ARM and LTM depend on the conditioning protocol. LTM is only acquired when spaced training protocols are used, whereas ARM is acquired after massed training. This is an important factor for how these two types of memory are formed, and it goes beyond energy reserves alone. See [Tully et al. (1994), Cell 79, 35-47] and [Isabel et al. (2004), Science 304, 1024-1027]. Currently, memory induction in the model presented here is the same for both ARM and LTM, and only depends on the energy reserve. Is it important that LTM and ARM are specifically modelled? Could ARM and short term memory (STM) be modelled instead, or ARM and medium term memory (MTM), or STM and MTM? Unfortunately, I do not know whether a similar difference in energy requirements hold for these other memory types. This would circumvent the difference with respect to spaced training, although ARM typically results from massed training of multiple associative exposures, whereas STM results from a single exposure, or at most a few exposures, but at least this would entail less of a difference in the conditioning protocol.

2) More recently, [Jacob and Waddell (2020), Neuron 106, 977-991] demonstrate that, in differential conditioning, LTM is acquired after spaced training, but is expressed as a bias to approach the CS-, and is encoded in a smaller avoidance MBON response to the CS- than to the CS+. This is in contrast to what one might expect: a bias to avoid the CS+, which would be encoded by a smaller approach MBON response to the CS+, as has been modelled in this manuscript. A parallel, decaying ARM memory, specific to the CS+, is encoded in the approach MBONs, but not in the avoidance MBONs. Thus, the formation of LTM in flies after spaced aversive conditioning involves a transition from a CS+ avoidance ARM to a CS- approach LTM. The paradigm used in this manuscript, however, is slightly different, in that there is no CS-, and it is not clear in Jacob and Waddell (2020) whether their results extend to stimuli that were not experienced during conditioning, i.e. whether all stimuli gain an approach LTM as for the CS-. Nevertheless, these results suggest that ARM and LTM are not mutually exclusive per se, but that both may result from plasticity at the inputs to different MBONs, which persist for different durations. More important, these results suggest that LTM formation entails ARM formation and decay, which is also supported by Isabel et al. (2004). Consequently, it is also important to discuss the suitability of using the same learning rate for ARM and LTM, as LTM forms after ARM. This needs to be addressed in the Methodology and in the Discussion, as the current implementation of the model does not capture these important details, and may be misleading to the naive reader.

3) Spaced training is addressed briefly in the Discussion, where it’s proposed that the fly should switch from ARM to LTM when the interval between conditioning exposures is increased. This is an appealing hypothesis. However, in experiments, the spacing interval is typically on the order of 15mins, which seems too short for the decay time of ARM, which is on the order of hours to days. A simulated experiment, with a smaller time step, to demonstrate how LTM has an advantage over ARM under spaced training protocols, would be an excellent addition to this manuscript, though I would not require it.

Minor comments:

1) Fig. 1C would benefit from additional explanation. I understand that stimuli further in the future have less impact on the expected reinforcements, as would be the case for discounting, but labelling the vertical axis as "Lifetime reduction" confuses me, because it includes that fact that the animal must live longer in order to see the stimulus further in the future. Some other confusing aspects that may help to clear this up include:

i) does the vertical axis convey the amount by which the lifetime is reduced (larger number implies greater reduction, with no upper limit), or the fraction of the lifetime that would be lived if the stimulus were not encountered (smaller number implies greater reduction)?

ii) Does the horizontal axis convey the duration over which the stimulus is experienced, or the time from now at which the stimulus will be next experienced?

iii) is this plot meant to be the survival function over time, i.e. the cumulative product S(t) = exp(-ht), where S is the survival probability, h is the hazard rate, and t is time?

2) Line 73-75 – I’m finding it difficult to understand these sentences. If M(t) is positive valued, then h_M is always 0. This doesn’t seem correct, unless I have misunderstood something. Could you please explain in more detail why you are using this new form for h_M, rather than that defined in Eq. 2.

3) Line 87-90 - Should references to Fig. 1b actually be Fig. 1c?

4) Line 87-90 - As per my previous comment on Fig. 1c, am I correct in thinking that the lifetime reduction is a prediction? As such, it makes sense that stimuli further into the future have less of an effect on predictions of lifetime reduction. It would help if this predictive nature were made explicit. If it is an absolute lifetime reduction, are we to interpret lifetime reduction as the integral of the lifetime reduction curve, up until the time at which the stimulus is presented?

5) Line 130 – Because Aso et al. (2014a) were not the first to discover that reinforcement is signalled by dopamine, I suggest a more appropriate reference, to guide those new to the field, would be [Waddell, S. (2013), Curr. Opin. Neurobiol. 23, 324–329] and references contained within.

6) Line 131 – I think Bennett et al. (2021) is not a suitable reference here, as it is a computational model, and does not provide evidence for DAN modulated plasticity, but rather implements that phenomenon in the model. Additional important references that would be good to include are [Hige, T. et al. (2015), Neuron 88, 985–998] and [Owald, D. et al. (2015), Neuron 86, 417–427].

7) Eqs. 6-7 – For those who wish to build upon this work, I think it would help to state that R-bar_+/- could, in general, be cue specific, and depend on w_+/-, even though only a single cue is modelled in this paradigm. If multiple cues were modelled, whereby the weights being updated depend on the sensory cue (Eq. 5), and R-bar_+/- were literally a sliding window average over reinforcements for all cues, delta_w +/- would be larger on average, which would affect the optimal thresholds, c_M and c_R, for ARM vs LTM memory.

8) Eqs. 6-7 – Following from point 8, if R-bar_+/- does not depend on the actual reinforcement predictions encoded in the weights, then delta_w will be larger than needs be for LTM. If R-bar did depend on LTM weights, then R-bar wouldn’t decay, and delta_w would be smaller on average for LTM formation, which in turn would affect the optimal c_M and c_R thresholds.. It would help to point this out explicitly, and to explain your choice model R-bar with a decay rate equal to that of ARM only.

9) Line 151 – Spurious “that” can be removed.

10) Line 162 - Please provide an explanation for why there's a difference between initialisation or ARM and LTM weights.

11) Figure 3a - 1) the different line thicknesses in the bottom panel, for hazards, are difficult to distinguish. Can a different dimension of line style be used, e.g. solid, dashes, dots? In particular, for LTM, I cannot see three lines for the M1 model. If one is obscured by another, please state this in the caption.

12) Figure 3a - why does the stimulus hazard decrease at the beginning of the simulation, when it should be fixed at 0.2?

13) Line 175 - It would be helpful if the stimulus protocol were described earlier, e.g. before the section "Reward driven plasticity", as this would help to provide a more tangible context in which to understand the description of w_+/- and R-bar updates.

14) Line 176-181 - The stimulus protocol, lasting 50 simulated days, seems at odds with the assumption made earlier that experiments are drastic such that age-related changes in the hazard can be ignored. That assumption makes sense when the model simulates an experiment lasting one day, but not a series of experiments lasting the lifetime of the fly. This choice of timescales needs more motivation.

15) Fig. 5 - Would help to state explicitly that it is the stimulus hazard only that takes low and high levels.

16) Line 232-233 – further clarification would help here by saying: “the larger the stimulus hazard, the lower the threshold that maximises lifetime.”

17) Line 241 – missing the word “in” between “a line” and “the M/DeltaR-plane”.

18) Line 257-259 – please show the data that support these these results.

19) Line 260 – I think references to Supplementary Fig. 9, here and elsewhere in the text, should be changed to Supplementary Fig. 1.

20) Line 278-279 – Please show the data.

21) The appendix in its current form is very difficult to follow, and would benefit from further explanation, suitable for a first year PhD student to follow. In particular, reintroducing symbols that were defined in the main text would be helpful. Some symbols have not been defined anywhere in the manuscript, e.g. lower case l, which I presume is lifetime, but had been defined using <t> beforehand.

22) Line 409 - it is claimed that the equation shows that a transition from LTM to ARM is preferable late in learning, when dP/dw is small, but this is opposite to what happens in reality, which is a transition from ARM to LTM. Please provide further clarification in the Appendix text.

23) Please provide instructions on Github to execute the code, including example commands.

24) Code appears to be missing functions that plot all of the figures, which would greatly promote this work's reproducibility.</t>

**Have the authors made all data and (if applicable) computational code underlying the findings in their manuscript fully available?**

Reviewer #1: Yes

Reviewer #2: Yes

Reviewer #3: **No: **Although much of the code is made available, there are no instructions to help others execute the code, and some code to plot the results appears to be missing. This things would greatly benefit this work's reproducibility.

PLOS authors have the option to publish the peer review history of their article (what does this mean?). If published, this will include your full peer review and any attached files.

Reviewer #1: No

Reviewer #2: **Yes: **Pierre-Yves Plaçais

Reviewer #3: **Yes: **James Bennett
---

## [Decision Letter · Decision Letter 1]

7 Oct 2024

Dear Dr. van Rossum,

Thank you very much for submitting your manuscript "Reinforcement learning when your life depends on it: a neuro-economic theory of learning" for consideration at PLOS Computational Biology. As with all papers reviewed by the journal, your manuscript was reviewed by members of the editorial board and by several independent reviewers. The reviewers appreciated the attention to an important topic. Based on the reviews, we are likely to accept this manuscript for publication, providing that you can make the minor modifications and corrections to the manuscript mentioned in the review recommendations.

Sincerely,

Barbara Webb

Academic Editor

PLOS Computational Biology

Daniele Marinazzo

Section Editor

PLOS Computational Biology

Reviewer's Responses to Questions

**Comments to the Authors:**

Reviewer #1: The authors have addressed my comments, and I recommend the article for acceptance.

Reviewer #2: The authors have done a very satisfactory revision of their already interesting manuscript. New explorations are presented when some of the assumptions or parameters are changed. In particular, it is extremely nice to see that the advantage of ARM and LTM ove lifetime vary oppositely when their learning rate is decreased.

The appetitive section has been largely improved and now shows that learning of large rewards can have a beneficial effect on lifespan, with relative memory dynamics that depend on the stimulus period.

There are a few remaining mistakes to correct :

- Figure 3b : the text in the legend box is duplicated.

- Figure 8a : No y-axis label ; legend box should not overlap with graphs. It would be better also to have separate x-axis labels.

Congratulations again to the authors for this nice and original of piece of science.

Reviewer #3: Many thanks for your hard work in addressing my previous comments. There remains a few minor suggestions for improvements below (line numbers correspond to the section where changes are highlighted):

1) Line 79: “The result is a steeper transition…” - this appears to come before ARM to LTM transitions are discussed, begging the question: steeper than what?

2) Line 150: The M1 energy model is referred to before being introduced. I suggest moving these two sentences about different learning rates to a later part of the paper, and replace it with a quick mention that different learning rates will be discussed later.

3) Fig. 1A: bottom panels need a legend to explain line thicknesses.

4) Line 303 and later: Could the word “consumption” be replaced with “expenditure” or something similar? I initially found this confusing, as “consumption” can have the same meaning as “intake”, and suggested to me that energy was gained, e.g. from the consumption of food.

5) It is interesting that the M1-energy model generally outperforms the M0-energy model when LTM is used exclusively, but when the adaptive threshold model is used, the M0-energy model outperforms the M1-energy model (comparing Fig. 6 with Supp. Fig. 5, and comparing Fig. 7 with Supp. Fig. 6). Are the authors able to comment on this and/or explain why, e.g. in the section “General threshold models”?

**Have the authors made all data and (if applicable) computational code underlying the findings in their manuscript fully available?**

Reviewer #1: Yes

Reviewer #2: None

Reviewer #3: Yes

PLOS authors have the option to publish the peer review history of their article (what does this mean?). If published, this will include your full peer review and any attached files.

Reviewer #1: No

Reviewer #2: **Yes: **Pierre-Yves Plaçais

Reviewer #3: **Yes: **James Bennett

Figure Files:

Data Requirements:

Reproducibility:

References:

---

## [Editor Report · Decision Letter 2]

14 Oct 2024

Dear Dr. van Rossum,

We are pleased to inform you that your manuscript 'Reinforcement learning when your life depends on it: a neuro-economic theory of learning' has been provisionally accepted for publication in PLOS Computational Biology.

Best regards,

Barbara Webb

Academic Editor

PLOS Computational Biology

Daniele Marinazzo

Section Editor

PLOS Computational Biology

---

## [Editor Report · Acceptance letter]

23 Oct 2024

PCOMPBIOL-D-24-00694R2 

Reinforcement learning when your life depends on it: a neuro-economic theory of learning

Dear Dr van Rossum,

I am pleased to inform you that your manuscript has been formally accepted for publication in PLOS Computational Biology. Your manuscript is now with our production department and you will be notified of the publication date in due course.

With kind regards,

Dorothy Lannert
